# Highly Stable, Mechanically Enhanced, and Easy-to-Collect Sodium Alginate/NZVI-rGO Gel Beads for Efficient Removal of Cr(VI)

**DOI:** 10.3390/polym15183764

**Published:** 2023-09-14

**Authors:** Qi Jing, Yuheng Ma, Jingwen He, Zhongyu Ren

**Affiliations:** Faculty of Architecture, Civil and Transportation Engineering, Beijing University of Technology, Beijing 100124, China; mayuheng@emails.bjut.edu.cn (Y.M.); a_wjbzd@163.com (J.H.); renzhy@bjut.edu.cn (Z.R.)

**Keywords:** gel bead, nanoscale zero-valent iron, reduced graphene oxide, sodium alginate, Cr(VI) removal

## Abstract

Nanoscale zero-valent iron (NZVI) is a material that is extensively applied for water pollution treatment, but its poor dispersibility, easy oxidation, and inconvenient collection limit its application. To overcome these drawbacks and limit secondary contamination of nanomaterials, we confine NZVI supported by reduced graphene oxide (rGO) in the scaffold of sodium alginate (SA) gel beads (SA/NZVI-rGO). Scanning electron microscopy showed that the NZVI was uniformly dispersed in the gel beads. Fourier transform infrared spectroscopy demonstrated that the hydrogen bonding and conjugation between SA and rGO allowed the NZVI-rGO to be successfully embedded in SA. Furthermore, the mechanical strength, swelling resistance, and Cr(VI) removal capacity of SA/NZVI-rGO were enhanced by optimizing the ratio of NZVI and rGO. Interestingly, cation exchange may drive Cr(VI) removal above 82% over a wide pH range. In the complex environment of actual Cr(VI) wastewater, Cr(VI) removal efficiency still reached 70.25%. Pseudo-first-order kinetics and Langmuir adsorption isotherm are preferred to explain the removal process. The mechanism of Cr(VI) removal by SA/NZVI-rGO is dominated by reduction and adsorption. The sustainable removal of Cr(VI) by packed columns could be well fitted by the Thomas, Adams–Bohart, and Yoon–Nelson models, and importantly, the gel beads maintained integrity during the prolonged removal. These results will contribute significant insights into the practical application of SA/NZVI-rGO beads for the Cr(VI) removal in aqueous environments.

## 1. Introduction

Water pollution caused by heavy metals is a critical issue since their detrimental effects on the environment and living organisms. Chromium is a typical heavy metal element commonly occurring in industrial wastewater, such as electroplating, textile printing and dyeing, steel and leather manufacturing, and accumulating in soil and water [1]. The toxicity of chromium is highly correlated with its valence, and hexavalent chromium (Cr(VI)), which is highly toxic relative to trivalent chromium (Cr(III)), is even more hazardous [2]. Since Cr(VI) is strongly oxidizing and water-soluble, long periods of exposure to high levels of Cr(VI) in humans may cause cancer and cellular tissue malformations [3]; therefore, there is a necessity to remove Cr(VI) from the aqueous environment.

Nano-zero-valent iron (NZVI) has been extensively applied to remove pollutants from groundwater, including heavy metals [4,5,6,7] and organics [7,8,9], because of its large specific surface area, high reactivity [10], and the possibility of injection into aquifers [11]. However, if NZVI is present at higher concentrations, NZVI tends to aggregate into larger particles owing to magnetic and van der Waals forces, thus reducing the specific surface area and mobility in water [12]. Moreover, under natural conditions, the structure and chemical properties of NZVI change due to its high reactivity, gradually oxidizing and corroding with time [13]. To enhance the dispersion and stability of NZVI, loading NZVI on porous materials is an effective approach. Currently, NZVI has been successfully loaded onto graphene [14], silica [15], carbon nanotubes [16], activated carbon [17], bentonite [18], and other materials. Loading NZVI onto graphene can effectively reduce particle agglomeration and show efficient treatment efficiency for organic matter [19] and heavy metals [20].

However, the practical applications of such nanocomposites are still restricted for reasons of economy and safety. Since most of these composites are nano-powder materials, they are difficult to separate and collect after application in water pollution remediation. From an economic point of view, the fact that inconvenient collection and reuse will undoubtedly cause an increase in treatment costs. What is more difficult is that from the perspective of environment and biomedicine, these nanomaterials that are inconvenient to collect are likely to cause secondary pollution to the environment. It has been reported that nano-scale materials, such as NZVI and GO are harmful to plants, animals, and even humans [21,22]. Therefore, the development of a remediation agent that is easy to collect, safe, and economical has become a hot topic of current research.

Alginate, a natural polysaccharide derived from seaweed, is a biodegradable, non-toxic, and water-insoluble gel gent [23]. Alginate can be chelated with divalent cations to form hydrogels, and alginate–calcium hydrogel/bead embedding is commonly used in the food industry and biomedical field for purposes such as the immobilization of living cells [24] and transportation of drugs [25]. In addition, the alginate-encapsulated NZVI technology has been applied in water treatment technology and the encapsulation does not affect the reactivity of NZVI. Bezbaruah et al. found that the reactivity of alginate-encapsulated NZVI for nitrate removal was comparable to that of bare NZVI and that entrapped NZVI could overcome the fluidity and sedimentation challenges related to bare NZVI [26]. The encapsulation process improves the co-removal efficiency of Cu(II) and MCB compared to Ni/Fe nanoparticles alone, and the gel beads can be reused multiple times with little effect on the removal efficiency, indicating the positive effect of encapsulation on degradation and reduction as well as improved economics [27]. However, alginate gels inevitably swell under certain conditions, which reduces their mechanical properties and limits the use of hydrogels [28]. It has been shown that the mechanical properties of sodium alginate (SA) beads can be improved by encapsulating graphene oxide (GO) in SA beads and then reducing them to form a reduced graphene oxide/sodium alginate (rGO/SA) bead [29]. Hence, the formation of gel beads by coating NZVI and rGO with sodium alginate not only avoids the disadvantages of nano-powder materials but also sufficiently utilizes the advantage that carbon materials can overcome the poor mechanical properties of sodium alginate.

Considering that rGO can not only improve the dispersibility and lifetime of NZVI but also enhance the mechanical strength of sodium alginate. Therefore, in this study, pre-synthesized NZVI supported by rGO (NZVI-rGO) was mixed with SA, and then the mixture was pumped into calcium chloride (CaCl_2_) solution to form the final product SA/NZVI-rGO gel beads. We demonstrate that SA/NZVI-rGO gel beads improve durability, mechanical strength, and NZVI dispersibility. Further, the ratio of NZVI-rGO was optimized to boost the anti-swelling and mechanical strength of SA/NZVI-rGO gel beads. Considering Cr(VI) as the target pollutant, the effects of bead composition, dose ratio of NZVI and rGO, pH, initial concentration of Cr(VI), and solid–liquid ratio on the removal effect were discussed. The kinetics, adsorption isotherms, and possible removal mechanisms of Cr(VI) removal by SA/NZVI-rGO gel beads were investigated. Finally, the performance of the gel beads under continuous removal was investigated by fixed-bed column experiments and simulated using three different models at different flow rates, concentrations, and column heights.

## 2. Materials and Methods

### 2.1. Materials

Sodium alginate (SA) and potassium dichromate (K_2_Cr_2_O_7_) were obtained from Tianjin Fuchen Chemical Reagent Co. Ltd., Tianjin, China. Sodium borohydride (NaBH_4_), diphenyl carbamide, anhydrous calcium chloride (CaCl_2_), and ferrous sulfate heptahydrate (FeSO_4_·7H_2_O) were bought from the Sinopharm Group Chemical Reagent Co. Ltd., Shanghai, China. Graphene oxide powder was received from Suzhou Hengqiu Technology Co. Ltd., Suzhou, China. None of the chemicals involved in the experiment were further purified. All solutions were made using deionized water.

### 2.2. Procedures for SA/NZVI-rGO Beads Synthesis

To load NZVI onto rGO (Figure 1a), GO (5 mg/mL) dissolved in water was pulverized by an ultrasonic cell pulverizer (Biosafer, 900-92, Biosafer Co., Nanjing, China) for 10 min, and then sonicated for 1 h in a water bath. Then, the GO suspension mixed with a solution of FeSO_4_·7H_2_O (2.482 g/ 20 mL) was transferred to a three-necked flask, and NaBH_4_ solution (2.5133 g/ 50 mL) was then dripped into the three-necked flask using a peristaltic pump at a rate of 1.5 mL/min. The mixture was constantly stirred and introduced with nitrogen gas. With the intense reductivity of NaBH_4_, Fe^2+^ and GO were reduced to NZVI-rGO. The final synthesized material was filtered through a 0.45 μm filter membrane and freeze-dried for 24 h. The NZVI-rGO at a ratio of NZVI: rGO = 1:1 was made. Considering the influence of different dosing ratios of NZVI and rGO on the final product, NZVI-rGO were also produced in ratios of 2:1, 3:1, and 5:1. The composite materials were named as xNZVI-rGO (1NZVI-rGO, 2NZVI-rGO, 3NZVI-rGO, and 5NZVI-rGO), where the x was the ratio of NZVI and rGO.

Next, to encapsulate the NZVI-rGO in SA (Figure 1b), 3 g of SA was dispersed in 50 mL of purified water and stirred for 3 h, and then left to defoam. NZVI-rGO (0.8 g/40 mL) dissolved in purified water was treated with an ultrasonic cell crusher for 10 min. The mixed SA and NZVI-rGO solutions were subsequently stirred under nitrogen for 1 h and then left to defoam. To obtain SA/NZVI-rGO gel beads, the mixed SA and NZVI-rGO were slowly dropped into CaCl_2_ solution (500 mL 10 mg/mL) using a peristaltic pump (Figure 1c). SA/NZVI-rGO gel beads rinsed repeatedly with purified water were freeze-dried. The beads were washed several times with deionized water and then freeze-dried. Unless otherwise stated, SA/NZVI-rGO beads below refer to SA/2NZVI-rGO beads.

To compare the performance of different gel beads, SA, SA mixed NZVI, and SA mixed rGO solutions were, respectively, dropped into 500 mL of 10 mg/mL CaCl_2_ solution with mechanical stirring to obtain SA gel beads, SA/NZVI gel beads, and SA/rGO gel beads.

To fit the mechanical test apparatus, SA/xNZVI-rGO mixture was poured into PTFE molds and injected with CaCl_2_ to gelate for 48 h to make rectangles (10 mm × 10 mm × 15 mm).

### 2.3. Characterization and Analysis Methods

Scanning electron microscopy (SEM, JSM-7900 F, JEOL Ltd., Tokyo, Japan) and Fourier transform infrared spectroscopy (FTIR, IRAffinity-1S, Shimadzu, Tokyo, Japan) were applied to observe the surface morphology and analyze the functional groups. An X-ray photoelectron spectroscopy (XPS, ESCALAB 250Xi, Thermo Fisher, Warrington, UK) was applied to determine elemental composition and valence state of gel beads. The zeta potential of products is measured with zeta potential analyzer (90 Plus, Brookhaven, NY, USA) at 25 ℃. The concentration of Cr(VI) was quantified by UV-Vis spectrophotometer (2802S UV/VIS, Shanghai Unicosh Instruments Co., Shanghai, China). The compressive strength was tested using an electronic universal testing machine (CMT6103, MTS, Akron, OH, USA) with a compression rate of 0.5 mm min^−1^. The swelling ratio (SR%) and water content (WC%) are illustrated in Appendix A.

### 2.4. Batch Experiments and Column Experiments

The Cr(VI) removal batch experiments were conducted by different gel beads (gel beads prepared by coating different components with SA as well as SA/NZVI-rGO beads by NZVI and rGO in different dosing ratios) to compare the removal effect and further optimize the SA/NZVI-rGO gel beads. The removal of Cr(VI) (100 mL) by beads at differing conditions was compared by batch experiments. The removal experiments were all performed using a water bath thermostatic oscillator at 150 rad/min. Aliquots were obtained at specified intervals and filtered with a 0.22 μm filter membrane. The effects of pHs (from 3 to 11), initial Cr(VI) concentrations (from 10 to 400 mg/L), and solid-to-liquid ratios (from 0.5 to 5 mg/mL) on Cr(VI) removal were considered separately. The calculation of removal capacity and removal efficiency is shown in Text S2.

The practical applicability of SA/NZVI-rGO gel beads was verified by treating real Cr(VI)-containing wastewater, which was taken from a chemical factory in Beijing, China. The composition of the actual Cr(VI)-containing wastewater is shown in Appendix A. Cation and anion contents were detected using inductively coupled plasma mass spectrometry (ICP-MS 7800, Agilent, Beijing, China) and ion chromatography (ICS-1100, Thermo Dionex, Warrington, UK), respectively.

Column experiments were conducted in Plexiglas columns (2 cm inner diameter × 14 cm height), which were tightly packed with SA/NZVI-rGO beads in the middle and packed with 2 cm thick glass beads at the top and bottom to ensure uniform water inflow and to prevent leakage of SA/NZVI-rGO beads from the bottom of the column, with the effluent flowing through the column in an upward direction. The effluent was obtained from the sampling port at regular intervals and sample collection was stopped when effluent concentration was almost equivalent to influent concentration. The effects of different flow rates (0.7, 2, 4 mL/min), bed heights (4.5, 6.5, 8.5 cm), and influent concentrations of Cr(VI) (10, 30, 40 mg/L) at a temperature of 25 °C and pH 3 of the Cr(VI) solution were considered. The breakthrough curve was obtained by plotting *C*_t_/*C*_0_ versus time.

## 3. Results and Discussion

### 3.1. Characterization of SA/NZVI-rGO Beads

The digital photographs of SA/NZVI beads, SA/NZVI-rGO beads, SA beads, and freeze-dried SA/NZVI-rGO beads, respectively, were shown in Figure 1 and Appendix A. All the beads showed a sphere-like shape with a diameter of about 3 mm and the color ranged from transparent (SA beads) to black (SA/NZVI beads and SA/NZVI-rGO beads). The encapsulated structure facilitates an easy separation of the gel beads from the treated wastewater. A certain mass of SA/NZVI-rGO beads was freeze-dried and then weighed, and the water content was 96.61%, as calculated by Equation (S2). In addition, SA/NZVI beads’ color turned from black to brownish red due to the rapid oxidation of the highly reactive NZVI in a short period (Figure 1a,c). However, SA/NZVI-rGO beads did not change their color after 22 h of Cr(VI) removal (Figure 1d), and the color of the SA/NZVI-rGO beads remained unchanged after 226 days of sealed storage (Appendix A), which indicates that the mere SA coating of NZVI did not reduce the oxidation rate of NZVI, but the pre-loading of NZVI on rGO can reduce the oxidation rate of SA/NZVI-rGO beads, which is beneficial for preservation in practical applications. And previous studies have also shown that NZVI can be coupled with carbon materials to build galvanic cells, which could expedite the electron release of NZVI and prevent the formation of oxidation films [30,31].

The SEM images in Figure 2 and Appendix A show the morphology and structure of NZVI, NZVI-rGO, and SA/NZVI-rGO. Appendix A reveals that massive amounts of NZVI particles are tightly connected to form large chain-like aggregates with poor dispersion. Figure 2a reveals the NZVI particles loaded on rGO nanosheets with folded lamellar structures are well dispersed. Our previous studies have demonstrated that well-dispersed NZVI could improve antioxidant properties [32]. These observations imply that NZVI with enhanced dispersion by rGO will contribute to increased reactive sites for the removal of contaminants. Figure 2b shows that SA/NZVI-rGO beads have a continuous three-dimensional staggered fibrous structure, and NZVI is well dispersed in beads, which indicates that SA/NZVI-rGO beads are more abundant in folds and channels than NZVI-rGO and greatly enhance the adsorption capacity of heavy metal contaminants.

FTIR spectra of SA beads, NZVI-rGO powder, and SA/ NZVI-rGO beads before and after Cr(VI) removal are compared in Figure 3. For SA beads, the wide band around 3400 cm^−1^ represented the telescoping vibrations of O-H, C-H at 2922 cm^−1^, O=C-O at 1430 cm^−1^, C=O at 1625 cm^−1^, and O-H at 1029 cm^−1^ represent the typical structure of alginate [33]. In NZVI-rGO, there is also a strong and wide peak near 3400 cm^−1^ associated with the telescoping vibration of the O-H group, with representative absorption peaks at 1648, 1398, 1128, and 1029 cm^−1^ ascribed to the deformation vibration peak of aromatic C=C, carboxy O=C-O, epoxy C-O, and hydroxyl O-H, respectively [34,35,36]. The weak peak at 692 cm^−1^ may be the Fe-O telescoping vibration belonging to Fe_3_O_4_ nanoparticles [37], which is consistent with the XPS results. Additionally, the spectra of NZVI-rGO and SA/NZVI-rGO remain almost identical, which indicates that SA successfully encapsulates NZVI-rGO. Specifically, the O=C-O of the SA/NZVI-rGO spectrum shifted significantly from 1398 cm^−1^ to 1428 cm^−1^ compared with the NZVI-rGO spectrum, indicating that some interfacial hydrogen bonds were formed between rGO and SA, which could cause a tight adhesion between the SA and rGO surfaces [38]. Meanwhile, the carbon atoms in rGO are bonded with sp^2^ hybrid orbitals, and the remaining electrons in the p orbitals can form delocalized π bonds. The O-H on the SA can contribute electrons to the free-moving π-electrons, thereby enhancing the π-electron cloud densities of the conjugated systems, which in turn promotes the tight adhesion of rGO on the SA [39]. These results indicate that NZVI-rGO can be uniformly and stably embedded in SA. The FTIR spectra of SA/NZVI-rGO before and after the removal of Cr(VI) (Figure 3) were utilized to identify the functional groups participating in the Cr(VI) removal. After Cr(VI) removal, a broad peak of O-H at 3400 cm^−1^ was observed to be narrowed, suggesting that O-H participated in the removal of Cr(VI). Also, an enhancement of the vibrational peaks of carboxyl groups at 1428 cm^−1^ (O=C-O) and 1618 cm^−1^ (C=O) was noted, which is congruous with prior studies that hydroxyl groups could serve as an electron donator to reduce Cr(VI) by being oxidized to carboxyl groups [40,41].

XPS spectra identified the elemental formation and chemical valence of the SA/NZVI-rGO beads before and after the removal of Cr(VI). Previous research has demonstrated NZVI has a kernel-shell system with an Fe(0) kernel and an iron oxide shell [42]. As shown in the Fe 2p spectra of SA/NZVI-rGO beads (Figure 4b), Fe_2_O_3_ at 711.95 eV (Fe 2p3/2) and 725.52 eV (Fe 2p1/2), Fe_3_O_4_ at 710.31 eV (Fe 2p3/2) and 723.11 eV (Fe 2p1/2), and Fe(0) at 705.79 eV (Fe 2p3/2), which proves the successful loading of NZVI particles on rGO nanosheets [43]. In Figure 4b, after Cr(VI) removal, the peak area of Fe(III) was elevated by 21.86% and that of Fe(II) was lessened by 2.22%, and Fe(0) disappeared, which confirmed that oxidative conversion of Fe(0) and Fe(II) to Fe(III) during Cr(VI) reduction by SA/NZVI-rGO gel beads. From the O1s spectrum (Figure 4c), the peaks observed at 531.06 eV, 529.51 eV, and 529.43 eV are attributed to O-H, O=C/O=C-O, and Fe-O, separately [44]. The mol ratio of Fe-O went up from 15.49% to 19.75% after the removal of Cr(VI), further demonstrating that the Fe(0) that provided electrons to Cr(VI) was being oxidized. Additionally, the mol ratio of O-H lessened from 66.43% to 52.84% and the molar ratio of O=C/O=C-O increased from 20.07% to 27.41%, which indicates that the hydroxyl groups donated electrons and were oxidized to carboxyl groups for the process of reducing Cr(VI) [45,46], which is consistent with the FTIR results. Cr 2p binding energy peak of Cr atoms was observed at 578.08 eV for the SA/NZVI-rGO beads after the removal in Figure 4a, and as shown in the Cr 2p split-peak spectrum (Figure 4d), Cr(III) corresponded at 576.03 eV (Cr 2p3/2) and 585.39 eV (Cr 2p1/2), while Cr(VI) at 578.43 eV (Cr 2p3/2) and 587.79 eV (Cr 2p1/2), indicating that both Cr(VI) and Cr(III) being reduced were adsorbed on gel beads. The results show that Cr(III) accounts for 85.69% of the total Cr on SA/NZVI-rGO, which means that the reduction is the primary cause for Cr(VI) removal. We further found that the intensity of the Ca 2p peak decreased (Appendix A), which implies that part of the Cr(III) ions possibly displaces Ca^2+^ to form carboxylated metal complexes within the gel beads instead of releasing them directly into the aqueous solution, which is consistent with the previous findings [47].

The zeta potential is applied to analyze the electrostatic characteristics of the material. To further explore the removal mechanism, Figure 5a demonstrates the zeta potential of SA/NZVI-rGO at different pHs. It is observed that SA/NZVI-rGO is positively charged at a low pH and has a strong negative charge when the pH was higher. At a low pH, Cr(VI) exists primarily as HCrO_4_^−^ [43], and as the pH < zero point charge (pHzpc), the surface of SA/NZVI-rGO is electropositive because of the carboxylate protonation, thus benefiting SA/NZVI-rGO in an acidic environment for Cr(VI) retention. However, when the pH > pHzpc, the gel beads are electronegative on account of the deprotonation of functional groups, which causes an electrostatic repulsion against the negatively charged Cr_2_O_4_^2−^, thereby being unfavorable to the adsorption. These prove that electrostatic interactions have a major role in Cr(VI) adsorption on SA/NZVI-rGO.

To investigate the effect of the surface charge on the stability of gels, Figure 5b shows the zeta potential values of SA gel, NZVI powder, NZVI-rGO powder, SA/NZVI gel, and SA/ NZVI-rGO gel. The zeta potential of SA gel is −0.16 mV, the NZVI powder is 1 mV, and the NZVI-rGO powder is −15.90 mV; therefore, the NZVI-rGO powder contains more charge than the first two. And SA/ NZVI-rGO gel has a higher zeta potential value than SA/NZVI gel, probably because the stronger electrostatic repulsion between SA and NZVI-rGO makes the gel enhanced and forms a more stable gel [29].

### 3.2. Swelling and Mechanical Properties

For the mechanical tests, all the gels were made into rectangles, as shown in Appendix A. The picture of the gel before, during, and after compression show that the gel is elastic. The stress–strain curves and Young’s modulus of gels with different rGO contents are shown in Figure 6a and Figure 6b, respectively. For SA/NZVI-rGO gels, no yielding was exhibited in any of the stress–strain curves, indicating neither fracture strain nor fracture strength is generated during compression [48]. The gels with a higher rGO content exhibited a higher Young’s modulus, indicating better compressive properties. For SA/NZVI gels without rGO addition, yielding occurred at a strain of 19.05% (Figure 6a), when the stress was 10 KPa, indicating that the addition of rGO and the higher rGO content resulted in better mechanical properties of the gels. Other studies have also found improved mechanical properties after adding rGO to alginate-based gels [29,49]. In Figure 6c, different gel beads were immersed in deionized water at different pHs, and the experimental procedure revealed that the SA/5NZVI-rGO gel beads in deionized water at pH = 13 were the first to dissolve and break compared to other pHs. At the same pH, the lower the rGO content, the higher the swelling ratio, which is consistent with the change in the decrease in the mechanical strength, further demonstrating that the swelling behavior decreases the mechanical strength. The reason for this phenomenon may be that rGO nanosheets dispersed uniformly in SA can restrict the movement of alginate chains [50]. Meanwhile, FTIR demonstrated that the interfacial hydrogen bonding and conjugated electron interaction between rGO and SA could enhance the compact structure within the gel beads, thus hindering and weakening the diffusion of water molecules within the gel beads. Thus, SA/NZVI-rGO gel beads, which improve the mechanical strength by adding the appropriate amount of rGO and maintain excellent anti-swelling properties in a broad pH range, are ideal for practical applications in the treatment of Cr(VI) contaminated wastewater.

### 3.3. Bath Experiments

#### 3.3.1. Effect of Different Bead Components

The prepared SA/NZVI-rGO gel beads showed a strong performance in removing Cr(VI), which is shown in Figure 7a. Cr(VI) at an initial concentration of 10 mg/L was removed with 0.5 g of SA/NZVI-rGO gel beads. The Cr(VI) removal reached equilibrium at the 80th minute, in which 98% of Cr(VI) was removed. To compare the contribution of the three components (SA, NZVI, and rGO) within the beads to the Cr(VI) removal, SA beads, SA/NZVI beads, and SA/rGO beads were made, respectively. As shown in Figure 7a, SA beads removed only about 1% of Cr(VI), SA/rGO beads improved the removal efficiency by about 14% over SA beads, whereas SA/NZVI beads improved the removal of Cr(VI) by about 95% over SA beads and by about 81% over SA/rGO beads. The SA/NZVI beads showed a relatively lower removal efficiency but a relatively higher removal rate than the SA/NZVI-rGO beads. In the gel beads, the NZVI acts as the active component to reduce Cr(VI) to Cr(III), and its high reactivity leads to a quicker and larger removal of Cr(VI). Nevertheless, compared to SA/NZVI-rGO gel beads, SA/NZVI gel beads are more prone to oxidation and show poor mechanical properties, which would limit the practical application of SA/NZVI beads. Therefore, we chose a SA/NZVI-rGO gel bead with oxidation resistance, enhanced mechanical properties, and effective Cr(VI) removal as the remediation agent.

#### 3.3.2. Effect of Different Dosing Ratios of NZVI and rGO

NZVI, which has strong reducibility, is the active ingredient in the reduction of Cr(VI) to Cr(III), and rGO is the component that can improve the mechanical properties of beads. To consider the influence of the dose ratio of NZVI and rGO on the removal capacity, 0.5 g SA/xNZVI-rGO gel beads were used to remove the Cr(VI) solution (100 mL, 10 mg/L). The removal capacity of the SA/NZVI-rGO beads for Cr(VI) increased with the dosage ratio of NZVI and rGO increasing (Figure 7b). The removal capacity of SA/2NZVI-rGO beads for Cr(VI) (2.07 mg/g) was almost twice that of SA/1NZVI-rGO beads (1.24 mg/g), which may be accounted for the more available active sites. However, as the dosing ratios of NZVI and rGO were further increased, SA/3NZVI-rGO beads and SA/5NZVI-rGO beads no longer showed a significant increase in the Cr(VI) removal capacity compared with SA/2NZVI-rGO beads, and the reaction rates of the three beads were similar in the early stage of Cr(VI) removal. The cause may be that the excess NZVI particles are severely agglomerated during the loading of rGO, thereby reducing the active sites, resulting in a no longer evident improvement in the removal capacity of Cr(VI) [51]. Also, considering the effect of rGO dosage on the mechanical properties and swelling ratio of the gel beads, SA/2NZVI-rGO was finally selected as the gel beads used in the subsequent experiments. The SA/2NZVI-rGO gel beads were used for reusability experiments, details and results are available in Text S3.

#### 3.3.3. Effect of Initial Solution pH

The different initial pH of solution also influences the Cr(VI) removal. As demonstrated in Figure 8a, with an initial pH of 3, the removal efficiency of Cr(VI) at the 80th minute could reach over 98%. As the pH increases to 5 and 7, the removal of Cr(VI) dropped to 91.74% and 89.67%, separately. When the solution was alkaline, the removal efficiency of Cr(VI) further dropped to 88.41% (pH 9) and 82.09% (pH 11). The decrease in the Cr(VI) removal efficiency with an increasing pH is mainly related to the charge on the gel beads and the species of Cr(VI) ions. On the one hand, SA/NZVI-rGO gel beads are susceptible to protonation in acidic solutions and have a strong electrostatic attraction to Cr(VI) anions. This can be confirmed by zeta potential of SA/NZVI-rGO gel beads at a different pH (Figure 5a). On the other hand, a different pH affects the Cr(VI) species, which influence the adsorption. The Cr(VI) species at different pH values were analyzed using Visual MINTEQ software (Figure 8b). At a low pH range, Cr(VI) was primarily in the form of HCrO_4_^−^ and when the pH increases it mainly appears as CrO_4_^2−^. And the reduction of Cr(VI) to Cr(III) is dependent on H^+^ ions in solution, as shown in Equations (1) and (2). HCrO_4_^−^ has less sorption-free energy compared to CrO_4_^2−^, which facilitates the adsorption [52,53].
(1)$$HCrO_{4}^{-}+7H^{+}+3e^{-}\to Cr^{3+}+4H_{2}O$$
(2)$$CrO_{4}^{2-}+8H^{+}+3e^{-}\to Cr^{3+}+4H_{2}O$$

However, the removal of Cr(VI) in the present study was almost pH-independent (even at pH = 11, Cr(VI) removal reached over 80%), which may be explained by the synergy of adsorption and reduction. Specifically, the reduction of Cr(VI) to Cr(III) is accompanied by the consumption of protons; however, it is noteworthy that with the increase in Cr(III), the cation exchange between Cr(III) and Ca(II) occurs on the three-dimensional structure of alginate, which is presented in the XPS results (Appendix A), and Ca(II) is re-released into environment, resulting in a localized pH decrease (hydrolysis process), which would again facilitate the adsorption and reduction. The minimum removal efficiency of SA/NZVI-rGO gel beads for Cr(VI) at a broad pH range was compared with other NZVI-based materials, and the results are presented in Appendix A. The results show that the SA/NZVI-rGO gel beads can remove Cr(VI) efficiently in a broad pH range.

#### 3.3.4. Effect of Initial Cr(VI) Concentration

Figure 8c shows the removal capacity and removal efficiency of Cr(VI) by SA/NZVI-rGO beads under different initial concentrations (10 mg/L–400 mg/L). The removal capacity of the SA/NZVI-rGO beads exhibited an increasing tendency with an increasing initial concentration, and the removal capacity of SA/NZVI-rGO beads reached a maximum of 55.42 mg/g at the initial concentration of Cr(VI) of 400 mg/L. When the removal reached equilibrium, the removal efficiency of Cr(VI) reached about 99% (at the initial concentration of 10 mg/L–100 mg/L), but as the initial concentration was further increased, the removal efficiency of Cr(VI) showed a decreasing trend, and the removal efficiency dropped to 70% at the initial concentration of 400 mg/L. When the concentration of Cr(VI) is high, a Fe-Cr hydroxide layer forms on gel beads to surround NZVI, thus preventing the reduction of Cr(VI) by electron transfer from NZVI [54].

#### 3.3.5. Effect of Solid-to-Liquid Ratio

The solid-to-liquid ratio between gel beads and Cr(VI) waste solution is also a key factor affecting the removal of Cr(VI). The Cr(VI) (100 mL) removal increased continuously (from 45% to 98%) with an increasing solid-to-liquid ratio (from 0.5 mg/mL to 5 mg/mL), while the removal capacity showed the opposite trend, decreasing from 9.57 mg/g to 2.07 mg/g, as shown in Figure 8d. It is attributed to a large amount of Cr(VI) competing for the available sites of gel beads at low dosing levels, resulting in a low removal efficiency but high removal capacity. As the dosage of SA/NZVI-rGO beads was appropriately increasing (from 0.5 mg/mL to 3 mg/mL), the increase in the Cr(VI) removal efficiency showed a rapid upward trend; however, as the dosage of gel beads was further increased (from 3 mg/mL to 5 mg/mL), the removal efficiency of Cr(VI) no longer increased significantly. The reason for this is that the higher the solid–liquid ratio, the more overlapping active sites there are [55].

#### 3.3.6. Cr(VI) Removal from Real Wastewater

The performance of the SA/NZVI-rGO gel beads for Cr(VI) removal from real wastewater was investigated. As shown in Figure 9, under the same operating conditions, when the removal reached equilibrium, the removal efficiency of the simulated Cr(VI)-containing wastewater reached 91.5%, while the removal efficiency of Cr(VI) in the actual wastewater reached 70.25%. As shown in Appendix A, the coexisting ions in real wastewater are usually quite complex. After the removal experiments, Ca^2+^ and Na^+^ in the wastewater increased due to the ion exchange of SA/NZVI-rGO gel beads, and most of the remaining anion and cation concentrations decreased. This proves that substantial coexisting anions and cations in the actual Cr(VI)-containing wastewater will vie with Cr(VI) ions for available sorption sites, leading to a decreased Cr(VI) removal. However, despite the complexity of the actual wastewater environment, the SA/NZVI-rGO beads still showed a nice Cr(VI) removal efficiency.

### 3.4. Kinetics Study of the Reaction

Based on the above results, the Cr(VI) removal by the SA/NZVI-rGO gel beads involves both adsorption and reduction. The adsorption kinetics models and reduction kinetics models were fitted to the experimental data for clarifying detailed kinetics of Cr(VI) removal.

#### 3.4.1. Adsorption Kinetics

Pseudo-first-order adsorption kinetics model and pseudo-second-order adsorption kinetics model are usually fitted for adsorption kinetics [56]. The pseudo-first-order adsorption kinetics equation and pseudo-second-order adsorption kinetics equation are below, respectively:(3)$$Q_{t}=Q_{e}(1-e^{-k_{1}t})$$
(4)$$Q_{t}=\frac{k_{2}Q_{e}^{2}t}{1+k_{2}Q_{e}t}$$
where *Q_e_* is the adsorption amount at equilibrium (mg/g); *Q_t_* is the adsorption amount at time *t* (mg/g); *k*_1_ is the rate constant of the pseudo-first-order adsorption kinetics model; *k*_2_ is the rate constant of the pseudo-second-order adsorption kinetics model.

According to the experimental data of Cr(VI) removal by the SA/xNZVI-rGO beads, the fitting results of the adsorption kinetics are presented in Table 1. As a result, correlation coefficients (R^2^) of pseudo-first-order adsorption kinetics are larger than that of pseudo-second-order adsorption kinetics, and the fitted *Q_e_* is closer to the measured *q_e_*, indicating that the Cr(VI) adsorption by SA/NZVI-rGO gel beads is more consistent with the pseudo-first-order adsorption kinetics, which demonstrates that physical sorption is the main rate control step in adsorption process of SA/NZVI-rGO gel beads [57].

#### 3.4.2. Reduction Kinetics

According to the experiments on the Cr(VI) removal by the SA/xNZVI-rGO beads, using the pseudo-first-order reduction kinetics model and the pseudo-second-order reduction kinetics model to fit the above experimental data [58]. The models of pseudo-first-order reduction kinetics and pseudo-second-order reduction kinetics can be described below:(5)$$\ln\frac{C_{t}}{C_{0}}=-k_{obs1}t$$
(6)$$\ln\left(\frac{1}{C_{t}}-\frac{1}{C_{0}}\right)=k_{obs2}t$$
where *C*_0_ is the initial contaminant concentration (mg/L); *C_t_* is the contaminant concentration at time *t* (mg/L); *k_obs_*_1_ is the pseudo-first-order rate constant (min^−1^); *k_obs_*_2_ is the pseudo-second-order rate constant (L/mg/min).

The pseudo-first-order/pseudo-second-order reduction kinetics parameters are listed in Table 2. The R^2^ of the pseudo-first-order kinetics fits ranged from 0.937 to 0.992; however, the R^2^ of pseudo-second-order kinetics fits were all less than 0.9, indicating that the Cr(VI) reduction by SA/NZVI-rGO gel beads is consistent with the pseudo-first-order reduction kinetics. It is in agreement with the statement that the pseudo-first-order kinetics model can correctly characterize the kinetics of NZVI particles and modified NZVI [59,60]. Additionally, when the ratio of NZVI to rGO was increased to 2:1, an abrupt increase in *k*_obs1_ was observed, indicating that SA/2NZVI-rGO has fast reduction kinetics. A comparison of *k*_1_ for the pseudo-first-order adsorption kinetics (Table 1) and *k*_obs1_ for the pseudo-first-order reduction kinetics (Table 2) revealed that adsorption served as the velocity-limiting step since the reduction process is far faster than that of adsorption [58].

### 3.5. Adsorption Isotherm Analysis

Further analysis of the adsorption mechanism was based on different initial concentration experiments. The experimental data were fitted using the Langmuir adsorption isotherm model and the Freundlich adsorption isotherm model. The Langmuir isotherm model hypothesizes that adsorption takes place on a monolayer where all adsorption sites are of equal energy, and the adsorption capacity is dependent on the number of adsorption sites on the adsorbent [61]. The linear form of Langmuir model could be written as follows:(7)$$\frac{C_{e}}{Q_{e}}=\frac{1}{k_{L}Q_{m}}+\frac{C_{e}}{Q_{m}}$$
where *C_e_* (mg/L) and *Q_e_* (mg/g) are the equilibrium concentration and equilibrium adsorption amount of Cr(VI) after removal; *k_L_* (L/mg) is a constant correlated with the adsorption energy; *Q_m_* (mg/g) is theoretical maximum adsorption amount.

Langmuir separation factor (*R_L_*) can reflect basic features of Langmuir isotherm, and *R_L_* determines whether the adsorption process is advantageous or not, and the equation is as follows:(8)$$R_{L}=\frac{1}{1+k_{L}C_{0}}$$
where *C*_0_ is initial concentration of adsorbent.

Freundlich isotherm model is empirical. It assumes that adsorbent has a non-homogeneous sorption surface with different binding forces at each adsorption site, and is multilayer adsorption [62]. The linear equation for the Langmuir model is represented below:(9)$$\ln Q_{e}=\ln k_{F}+\frac{\ln C_{e}}{n}$$
where *k_F_* is a constant relevant to the capacity of the adsorbent; *n* is a constant that responds to the sorption strength of the adsorbent.

The fitted isothermal adsorption parameters from the experimental data of different initial concentrations are shown in Table 3. The relevant coefficient of the Langmuir isothermal model (R^2^ = 0.968) is larger than that of the Freundlich isothermal model (R^2^ = 0.924), so the Langmuir isothermal adsorption model provides a more favorable fitting of the measured data. The *R_L_* was between 0 and 1 for this experiment, indicating that adsorption is easy to perform [63]. Furthermore, the maximum theorized capacity for Cr(VI) removal by SA/NZVI-rGO gel beads using the Langmuir isothermal adsorption model is 53.42 mg/g, which is approaching the measured maximum sorption capacity (55.42 mg/g). The removal capacity of SA/NZVI-rGO gel beads for Cr(VI) was contrasted with that of other NZVI-based materials, and the results are shown in Appendix A. It can be observed that the SA/NZVI-rGO gel beads have a comparable removal capacity. Although this capacity does not remarkably increase to other published NZVI-based materials, the amount of NZVI used in this study is relatively low (per gram of SA/NZVI-rGO beads contains only 0.14 g NZVI).

### 3.6. Removal Mechanism

Reasonable removal mechanisms, including reduction, electrostatic interaction, and ion exchange, are proposed in Figure 10. The Cr 2p XPS spectra and kinetic results confirmed Cr(VI) removal by gel beads was achieved by adsorption and reduction, which was the main rate control step. Specifically, the high water content (96.61%) of the SA/NZVI-rGO gel beads promoted the penetration of the Cr(VI) solution, and the abundant folds and channels within the gel beads shown by SEM favored the attachment of Cr(VI). Under different pHs, the zeta potential of SA/NZVI-rGO beads (Figure 5a) and Cr(VI) removal experiments (Figure 8a) showed that the electrostatic attraction between positively charged SA/NZVI-rGO gel beads and negatively charged Cr(VI) favored the adsorption of Cr(VI) by SA/NZVI-rGO in acidic environments. Furthermore, the XPS spectra of SA/NZVI-rGO before and after Cr(VI) removal demonstrated that Fe(0) was oxidized to Fe(II) and Fe(III), as well Cr(VI) was reduced to Cr(III) through obtaining electrons given by Fe(0), and Cr(III) reacts with OH^−^ in an aqueous solution to form Cr(OH)_3_ deposited on the gel beads. The presence of rGO led to a faster reduction of Cr(VI) to Cr(III) due to the formation of massive primary batteries by NZVI and rGO to accelerate the release of electrons [64,65]. Then, Fe(II), which is the result of the oxidation of Fe(0) losing electrons, can further reduce Cr(VI) to Cr(III), and Fe(II) was eventually oxidized to Fe(III). Additionally, FTIR (Figure 3) and O 1s XPS spectra (Figure 4c) of SA/NZVI-rGO before and after Cr(VI) removal demonstrate the hydroxyl group is an electron donor that can reduce Cr(VI) to Cr(III) and then to be oxidized to carboxyl groups. It is interesting to note that in the alginate matrix, Cr(III) can undergo an ion exchange with -COO...Ca...OOC- (Appendix A), thus releasing Ca(II) to further decrease the solution’s pH. This means that more H^+^ is available in the aqueous solution for electrostatic adsorption due to the protonation of functional groups and for reduction, which may indirectly improve the removal [66]. Thus, the synergistic combination of the proposed removal mechanisms can provide the SA/NZVI-rGO gel beads with a great removal capacity for the effective Cr(VI) removal from aqueous solutions.

### 3.7. Column Experiments and Model Fitting

Column experiments were applied to research the continuous Cr(VI) removal by SA/NZVI-rGO beads. Theoretical column models are available to simulate the breakthrough performance of columns packed with SA/NZVI-rGO beads, and the Thomas, Adams—Bohart, and Yoon—Nelson models were fitted and analyzed to the measured data, and the above models are described in detail in Text S4. As can be seen from Figure 11a,d,g, when the bed height rose from 4.5 cm to 8.5 cm (flow rate 0.7 mL/min, influent concentration 10 mg/L), the breakthrough time and removal performance increased due to the increase in the adsorption binding sites and the prolonged exposure time of Cr(VI) to the gel beads, resulting in an improved removal efficiency of Cr(VI) [47]. As a result, the higher filling of the column would slowly decrease the removal performance of the column, which is ideal for operation, and a similar finding has been published in the literature [67]. When the influent concentration of Cr(VI) was increased from 10 mg/mL to 40 mg/mL (flow rate 0.7 mL/min, bed height 8.5 cm) (Figure 11b,e,h), the breakthrough time and depletion time of gel beads bed will be advanced because the binding sites would saturate more quickly at higher concentrations [68]. When the flow rate was accelerated from 0.7 mL/min to 4.0 mL/min (bed height 4.5 cm, influent concentration 10 mg/L) (Figure 11c,f,i), both the exhaustion time and removal performance decreased, and the faster flow rate reduced the contact time of Cr(VI) with SA/NZVI-rGO beads, resulting in a steeper penetration curve. This is because a too fast flow rate increases the mass transfer rate, which leads to the increase in the Cr(VI) adsorbed per unit bed height and faster saturation of the bed column, which is consistent with the literature reports [69,70]. Furthermore, due to the excellent mechanical properties and enhanced swelling resistance of the SA/NZVI-rGO gel beads, the gel beads in fixed-bed column keep their integrity after continuous removal. NZVI-rGO confinement in SA reduces nanomaterial contamination due to uncontrolled migration of nanoparticles. The SA/NZVI-rGO gel beads are available for treatment of actual Cr(VI) wastewater.

As presented in Figure 11, these models were fitted to the measured data from column experiments under different conditions. The parameters of each model were acquired from the nonlinear form of the model equation and summarized in Appendix A. When the bed height increased, the flow rate increased and the influent concentration decreased, respectively, *K*_T_, *K*_AB,_ and *K*_YN_ increased. The well-fitted Thomas model suggested external diffusion and internal diffusion are not restrictive steps, and the well-fitted Adam–Bohart model demonstrates that external mass transfer governs dynamics of the whole system [67]. It can be concluded from the Yoon–Nelson model’ fit that τ increases with a higher bed height and a lower influent concentration and flow rate, respectively, and τ increases indicating a slower fixed bed exhaustion, which is ideal for removal processes. All models have high correlation coefficients (R^2^ > 0.823), indicating that all three models have a good fit, there is evidence that the above models, which are mathematically verified to be equivalent, all exhibit high levels of fit performance [71].

## 4. Conclusions

In this study, the NZVI that was pre-loaded on rGO was embedded into SA to synthesize SA/NZVI-rGO gel beads for removing Cr(VI) from the aqueous solution. The research proved the following: (1) The optimized gel beads simultaneously improved oxidation resistance, swell resistance, and compression strength. (2) The SA/NZVI-rGO gel beads could achieve a substantial Cr(VI) removal in a broad pH range (3–11) or a wide Cr(VI) concentration range (10–100 mg/L). The Langmuir adsorption isotherm demonstrated that the adsorption of Cr(VI) by SA/NZVI-rGO occurred easily, and the theoretical maximum removal capacity was 53.42 mg/g. (3) The removal mechanism of Cr(VI) is a synergy of adsorption, reduction, and ion exchange, where the reduction is dominant. (4) The Adams–Bohart, Thomas, and Yoon–Nelson models showed a good fit for the continuous Cr(VI) removal under different conditions in packed columns. Importantly, the gel beads in the column maintained integrity all the time, owing to their excellent mechanical strength, not only facilitating collection but also limiting the uncontrolled migration of NZVI-rGO, which provides a reliable basis for reducing secondary environmental contamination by nanomaterials in practical applications. Overall, by demonstrating improved oxidation resistance, enhanced mechanical properties and swelling resistance, easy collection, and efficient Cr(VI) removal, the SA/NZVI-rGO beads are a promising remediation agent in Cr(VI) wastewater treatment.

## Data Availability

The data that have been used are confidential.

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
