# Peer review of "Highly Stable, Mechanically Enhanced, and Easy-to-Collect Sodium Alginate/NZVI-rGO Gel Beads for Efficient Removal of Cr(VI)"

_polymers, 2023, doi:10.3390/polym15183764_

Round 1

Reviewer 1 Report

The present paper demonstrates the efficient adsorption properties of graphene oxide gel beads containing zero-valent iron species. I agree well with significance of the paper. I have one simple question on the adsorption mechanism for efficient Cr(VI) species. I recommend the explanation on this point for a deeper understanding of the present adsorbent. 

1. With regard to the adsorption mechanism of Cr(VI) removal by SA/NZVI-rGO, the valence of iron became +II or +III from zero, as explained in Figure 4b and 10. I am wondering whether you have another experimental evidence such as XRD patterns. Besides, when the Iron species are oxidized, the SA/NZVI-rGO shows no adsorption properties against Cr(VI) species? These two points are interesting for readers.

Author Response

Thank you very much for your question.
(1) Your comment is very enlightening to me, XRD can explore the crystal structure, chemical composition and morphology of materials.
In the study by Ai et al., XRD found that Fe0 was oxidized to Fe2+ to Fe3+ during aging Fe@Fe in water, and Fe2+ could be re-precipitated as FeO, Fe3O3 and FeOOH by reacting with components in water.
Fe3O4 and FeOOH react with components in water [3]. In future work, we will also apply XRD to analyze changes in the valence state of iron.
(4) The mechanism of Cr(VI) removal by SA/NZVI-rGO is comprehensive and synergistic, including not only reduction of zero-valent iron, but also electrostatic attraction, ion exchange, and electron donation by hydroxyl groups to reduce Cr(VI).
To verify Cr(VI) removal by other mechanisms, we performed reusability experiments with SA/NZVI-rGO gel beads (in Supplementary Material Text S1), and in four applications, we found that even multiple Application leads to iron oxidation, and SA/NZVI-rGO still has a certain amount of Cr(VI) removal.
quote Ai, Z.; Gao Zhiming; Zhang Li; He, W.; Yin, J.J. The core-shell structure of Fe@Fe2O3 nanowires depends on the reactivity of 4-chlorophenol for aerobic degradation. Environment. Science and Technology. 2013, 47, 5344–5352, doi: 10.1021/es4005202.

非常感谢您的提问。(1)您的评论对我很有启发,XRD可以探索材料的晶体结构,化学成分和形貌。在Ai等人的研究中,XRD发现Fe0在水中老化Fe@Fe过程中被氧化为Fe 2+到Fe 3+,Fe 2+可以通过与水中的成分反应而重新沉淀为FeO,Fe3O3和FeOOH。Fe3O4和FeOOH通过与水中的成分反应[3]。在今后的工作中,我们还将应用XRD来分析铁价态的变化。(4)SA/NZVI-rGO去除Cr(VI)的机理是全面和协同的,不仅包括还原零价铁,还包括静电吸引、离子交换和羟基提供电子来还原Cr(VI)。为了验证其他机制对Cr(VI)的去除,我们用SA / NZVI-rGO凝胶珠(在补充材料文本S1中)进行了可重复使用性实验,并且在四次应用中,我们发现即使多次应用导致铁氧化,SA / NZVI-rGO仍然存在一定量的Cr(VI)去除。

引用

  1. 艾,Z.;高志明;张立;他,W.;Yin, J.J. Fe@Fe2O3纳米线的核壳结构依赖于4-氯苯酚好氧降解的反应性。环境. 科学技术. 201347, 5344–5352, doi:10.1021/es4005202.

Reviewer 2 Report

The manuscript entitled "Highly stable, mechanically enhanced, and easy-to-collect sodium alginate/nano zero-valent iron-reduced graphene oxide gel beads for efficient removal of Cr (VI) presents an interesting experimental study on the obtaining and efficiency testing of NZVI in Cr (IV) removal. However, the paper has a few issues that must be addressed. The paper needs minor revisions before it is processed further. Some comments follow:

The title is too long. Could authors replace it with a shorter formula that clearly reflects the content of the paper but also attracts the interest of readers in this field?

Results section.

Please introduce the scale bar in Figure 1.

SEM images: please introduce figure labels to indicate the components of the analyzed surface.

Conclusion section: Please improve the conclusions and present them following the main recommendations by the academy of giving the conclusions of the study by points with highlights.

Future directions and limitations: Please provide some future directions and limitations of the study.

English grammar and spelling

The paper has multiple typing errors:

even more hazardous [2].

malformations. [3].  

Please remove the dot prior to citing.

humans[21, 22]. Space is missing.

On Page 5, after "heavy metal contaminants, there is a symbol that should be removed.

Round 2

Reviewer 1 Report

The present paper is well written with regard to the preparation and characterization of SA/NZVI-rGO.